# A simplified method for blood feeding, oral infection, and saliva collection of the dengue vector mosquitoes

**Chalida Sri-in[1]◉, Shih-Che Weng[2]◉, Shin-Hong Shiao📷[2]\*, Wu-Chun Tu📷[1]\***

**1** Department of Entomology, College of Agriculture and Natural Resources, National Chung Hsing University, Taichung, Taiwan, **2** Department of Tropical Medicine and Parasitology, College of Medicine, National Taiwan University, Taipei, Taiwan

◉ These authors contributed equally to this work.
\* wctu@dragon.nchu.edu.tw (WCT); shshiao@ntu.edu.tw (SHS)

**Data Availability Statement:** All relevant data are within the manuscript and its Supporting Information files.

## Abstract

A simple device using folded Parafilm-M as an artificial blood feeder was designed for studying two important dengue vector mosquitoes, *Aedes aegypti* and *Aedes albopictus*. The efficiency of the artificial blood feeder was investigated by comparing the numbers of engorged mosquitoes that fed on the artificial blood feeder *versus* mice as a live blood source. Significantly more engorged females *Aedes aegypti* fed on the artificial blood feeder than on mice. In addition, the artificial feeder could serve as a useful apparatus for oral infection via artificial blood meals, and for saliva collection in mosquitoes. Our method enabled us to collect saliva from multiple mosquitoes at once, providing sufficient infected saliva for determination of the virus titer by plaque assay analysis. Our artificial feeder has the advantage that it is simple, inexpensive, and efficient.

## Introduction

The dengue vector mosquitoes, *Aedes aegypti* and *Aedes albopictus*, require blood feeding for egg maturation [1]. Anesthetized or immobilized animals such as mice, other rodents, and lagomorphs are usually used as the main blood source for maintaining mosquitoes in insectaries [2–5]. However, under some circumstances, it is difficult to use live animals for feeding these vectors due to a need for ethical approval, restrictions by animal care committees, and many regulations in place for the use of live animals in experiments [2, 6–8]. In addition, using live animals is costly and maintaining them is laborious.

Artificial blood-feeding systems are important to maintain mosquito populations in the laboratory and can be applied to replace live animals as blood sources [2]. A number of methods have been developed to feed blood-sucking insects [6,9–11], most of which use animal skins or thin membranes filled with warmed animal blood, that allow female mosquitoes to insert their proboscis to acquire a blood meal. These devices are generally constructed using costly materials, and are complicated to assemble [12]. Additionally, to inoculate mosquitoes with dengue

**Funding:** This study was supported by research grant (MOST 107-2313-B-005-027) from Ministry of Science and Technology, Executive Yuan, ROC.

**Competing interests:** The BALB/c mice were bought from Lexco Biotechnology Co., Ltd. Therefore, this does not alter our adherence to PLOS ONE policies on sharing data and materials.

virus (DENV) for various experiments, an artificial blood-feeder system is usually required for mosquitoes to take up infectious blood meals [13–14].

Furthermore, DENV transmission can be determined directly by collecting infected mosquito saliva to examine the presence of virus. Several devices such as capillary tubes and suspended droplets for collecting mosquito saliva are available [15–21] but it is time-consuming to collect saliva from large numbers of mosquitoes, and the saliva collected does not contain sufficient virus titer for studying DENV transmission by means of plaque assay, or by using a mouse model to examine hemorrhage development. Therefore, we developed a simple Parafilm blood feeder which has many benefits including blood feeding, oral infection, and saliva collection of mosquitoes. This apparatus has the advantage over existing artificial feeder systems that it is simple, convenient, and low cost, and requires no power supply or heater.

## Materials and methods

### Mosquito rearing

The *Ae. aegypti* UGAL/Rockefeller strain and *Ae. albopictus* Kaohsiung strain were maintained until adult stage at 25 ± 1˚C, 80% relative humidity, and a 12:12h (Dark:Light) photoperiod as described previously [22]. Female mosquitoes 3 to 5 days after emergence were used for various experiments.

### Mice rearing

Mouse-use for blood supply tests has been reviewed and approved by the Institutional Animal Care and Use Committee (approval no. 102–76), effective from October 23, 2013 to October 17, 2018. BALB/c mice were bought from Lexco Biotechnology Co., Ltd. All mice were maintained under specific-pathogen-free conditions at the Medical Entomology Laboratory, National Chung Hsing University, Taiwan. The mice were housed in accordance with local and home office regulations. Three female mice, five-week-old and specific pathogen-free, were anaesthetized using Pentobarbital sodium injection with a fixed dose of 75 mg/kg i.p. during the blood feeding experiment for 30 min. The mice used in this experiment were not euthanized.

### Cell culture and dengue virus production

The *Ae. albopictus* C6/36 cells containing 2% heat-inactivated fetal bovine serum (FBS) and 1× penicillin–streptomycin solution were cultured in a mixture of Dulbecco's modified Eagle's medium (DMEM, Gibco) and Mitsuhashi and Maramorosch insect medium (MM) at 1:1 ratio. These cultured cells were infected with DENV2 strain 16681 at an infection multiplicity of 0.01. The culture supernatant was harvested 5 days after infection and subjected to a plaque assay to determine the viral titer. DENV2 at approximately $1.0 \times 10^7$ PFU/ml was used to infect the mosquitoes. The plaque assay method will be further explained below.

### Mosquito blood-feeding and egg laying experiments

Females of *Ae. aegypti* and *Ae. albopictus* were transferred into cylindrical containers fitted with nylon mesh on top and starved through sugar-deprivation for 24 h. Mosquitoes were subsequently offered a blood meal containing 350 µl of washed sheep erythrocytes, 100 µl of 2% FBS, and 50 µl of 1 mM adenosine triphosphate (ATP). The meal was wrapped in stretched Parafilm-M membrane and warmed at 37˚C, and then placed on the top of a container covered with nylon mesh (Fig 1A). Fifty mosquitoes inside the container were allowed to feed on the meal through the stretched Parafilm-M membrane (Fig 1B), or on anaesthetized mice as a

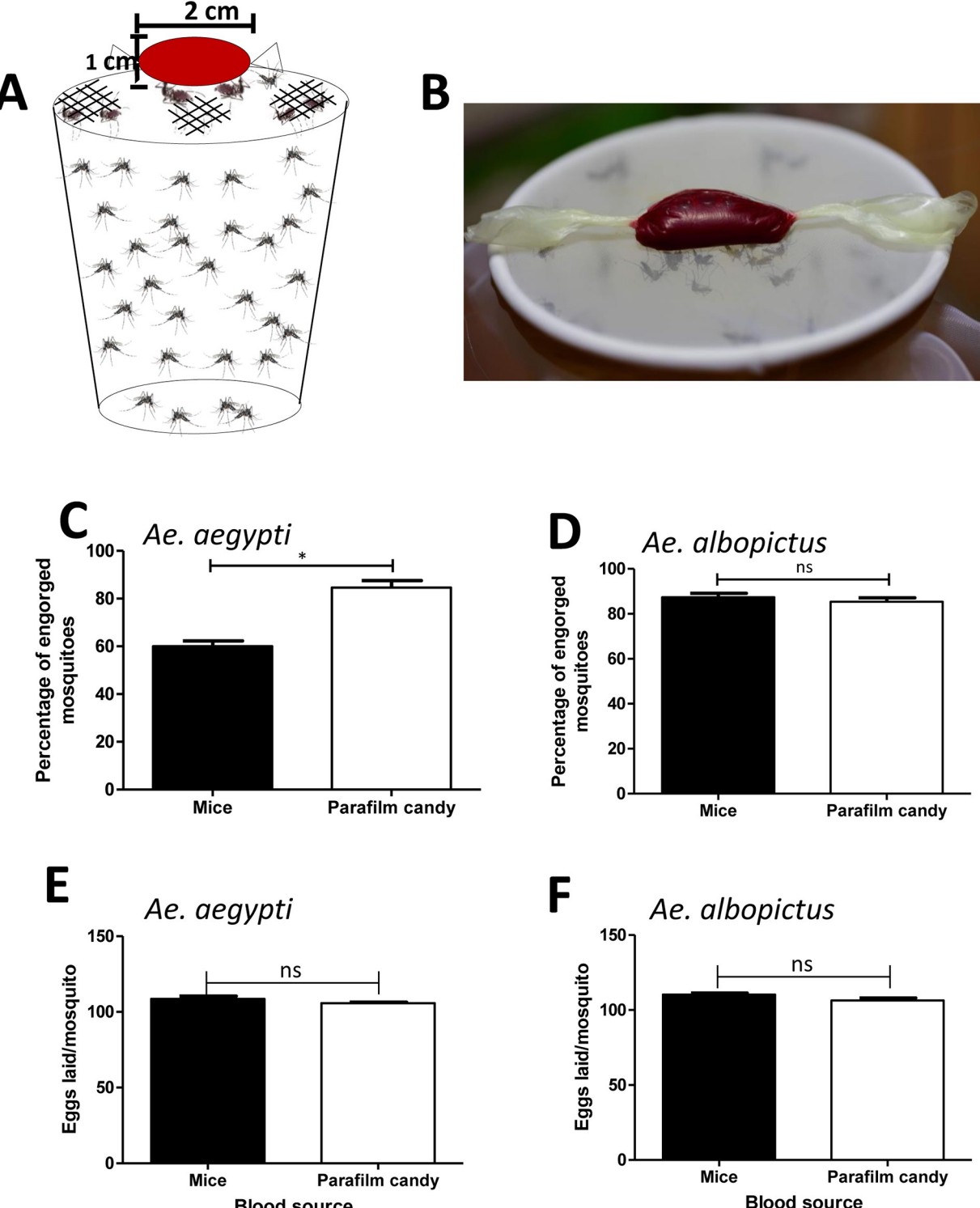

**Fig 1. Mosquito blood-feeding using artificial blood feeder. A**. Blood meal containing 100 µl of 2% FBS, 50 µl of ATP, and 350 µl of erythrocyte was wrapped in stretched Parafilm-M membrane and put it on the top of a container covered with nylon mesh. Female mosquitoes were allowed to take up blood meal by bite wrapped blood inside parafilm. **B**. Mosquitoes inside the container were allowed to feed on the blood meal through the artificial blood feeder. **C**. To generate blood-feeding efficiency, the numbers of engorged females after 30 min of feeding on mice or artificial blood feeder were counted. Percentages of successful feeding were analyzed according to the number of engorged females divided by the total number of females allowed to feed. Data were pooled from three independent experiments and presented as mean ± SEM. *$P < 0.05$ (Unpaired *t*-

test). **D**. To generate the effect of blood source on eggs laid, the numbers of eggs laid from engorged females that fed on mice or artificial blood feeder were counted. Each replication represented the eggs laid of 20 mosquitoes. Data were pooled from three independent experiments and presented as mean ± SEM. ns, not significant (Unpaired *t*-test).

control for 30 min. The surface of the mice available to the mosquitoes in the control group was limited to a 2 cm$^2$ hole, the same diameter as the artificial blood feeder offered to the mosquitoes. Each mosquito fed either on an artificial blood meal or on a mouse was examined on a stereo microscope (Leica EZ4 HD) to determine whether it had taken a full meal and to evaluate the percentage of engorged females that their abdomens were bloated with blood after approximately one minute of feeding.

All engorged females were placed in a container covered with nylon mesh with a cotton pad soaked with 10% glucose solution for feeding. On day 5 post blood meal, all engorged females were transferred and placed individually in 50-ml plastic tubes in which cotton soaked with mineral water was put on the bottom, and white paper was placed around the bottom for the females to lay eggs [23]. The oviposition tube was covered with a mesh, and a cotton pad soaked with 10% glucose solution was placed on top of each tube. The number of eggs laid by each female was counted.

## Oral infection of mosquitoes with DENV2

Infection of mosquitoes was achieved through an infectious blood meal. *Ae. aegypti* females were transferred into cylindrical containers fitted with nylon mesh on the top and starved through sugar deprivation for 24 h. Female mosquitoes were subsequently offered an infectious blood meal prepared by mixing 200 μl of washed sheep erythrocytes, 50 μl of 1 mM ATP, and 250 μl of DENV2 16681 (2.5 x 10$^6$ PFU in 250 μl) [24]. The meal was contained in a stretched Parafilm-M membrane (Fig 1A and 1B). The mosquitoes were allowed to take up the infectious blood for 1 h. Each mosquito was examined on a stereo microscope to determine whether it had taken a full meal. Mosquitoes were then transferred into cylindrical containers fitted with nylon mesh on the top and held in 10% glucose solution at 25˚C and 80% humidity, according to standard rearing conditions [25].

## Mosquito saliva collection and salivary protein detection

To collect saliva for measuring protein concentration, *Ae. aegypti* and *Ae. albopict*us females were starved for 24 h prior to saliva collection. On the day of saliva collection, feeding solution containing 90 μl of 1× phosphate-buffered saline (PBS) and 10 μl of 1 mM ATP at final concentration 10 μM [26–27] was prepared. The feeding solution was wrapped in stretched Parafilm-M membrane (Fig 2A and 2B) and put on the top of a container covered with nylon mesh, allowing 1, 10, 20, 50, 100, and 150 mosquitoes to feed on the meal (Fig 2B). The feeding solution containing mosquito saliva was removed from the membrane and transferred to a microtube and centrifuged at 12,000 *g* for 1 min at 4˚C. The saliva's protein concentrations were measured using Bradford protein assays.

To collect infected saliva for plaque assays, female mosquitoes were starved for 24 h prior to saliva collection. Infected mosquito saliva was examined on day 3, 5, 7, 10, and 14 post infectious blood meal. On the day of saliva collection, feeding solution containing 180 μl of 1× PBS and 20 μl of 1 mM ATP at a final concentration of 10 μM was prepared. The feeding solution was wrapped as described previously and put on the top of a container covered with nylon mesh, allowing 200 infected mosquitoes inside to feed on the meal (Fig 2A and 2B). The infected mosquitoes were allowed to feed for 30 min. The feeding solution containing mosquito saliva was removed from the membrane as described previously. This was repeated 5

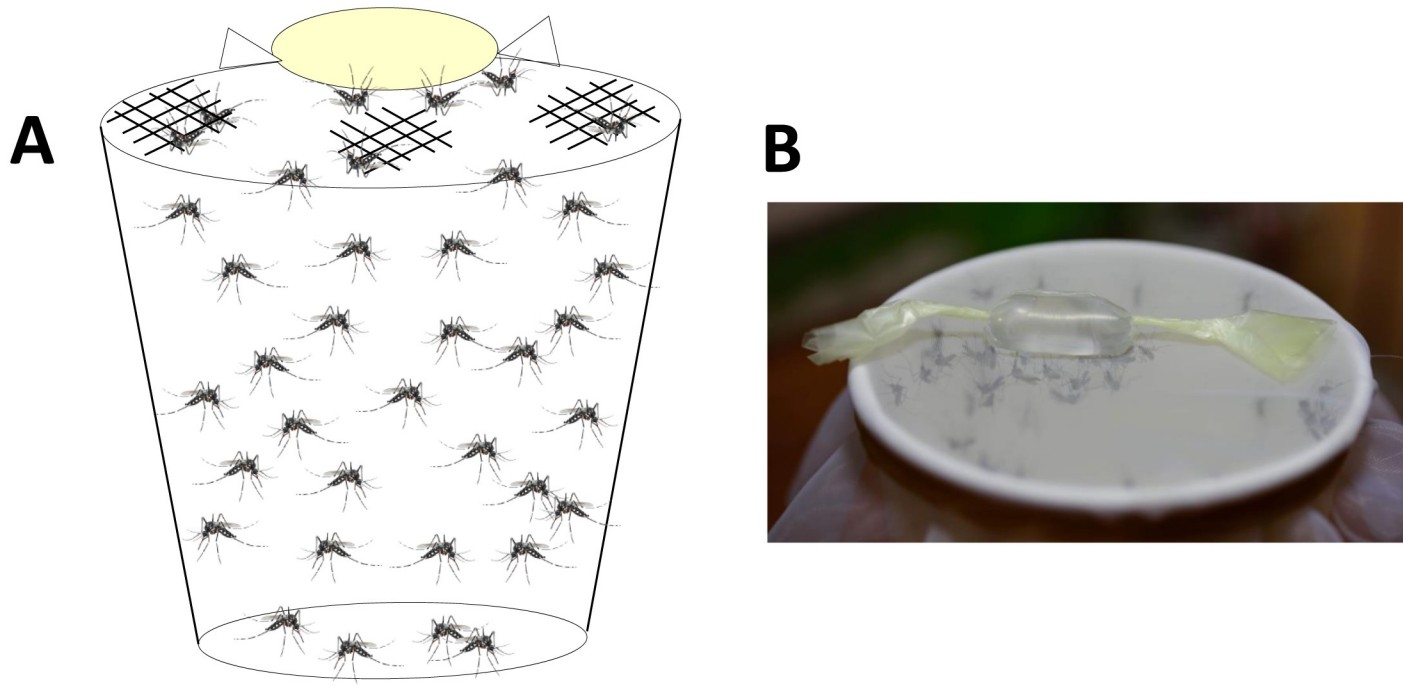

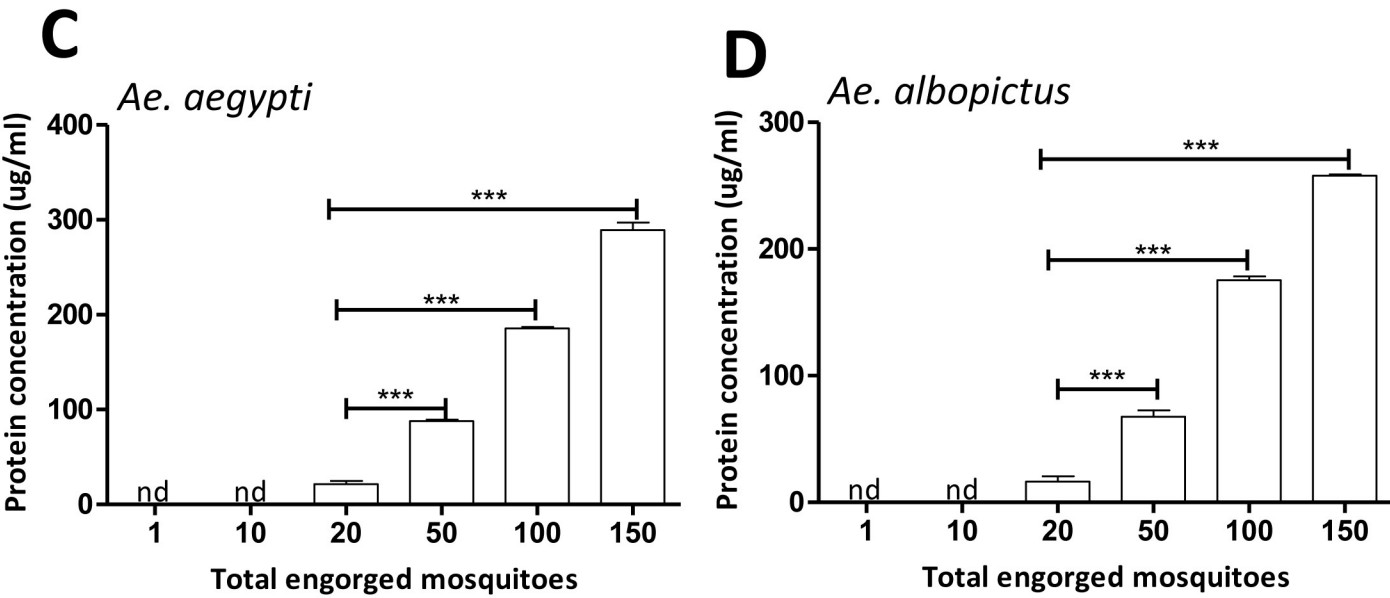

**Fig 2. Mosquito saliva collection using artificial blood feeder. A**. To collect mosquito saliva using artificial blood feeder, feeding solution containing 90 ul of 1× PBS and 10 ul of 1 mM ATP was wrapped in stretched Parafilm-M membrane and put on the top of a container covered with nylon mesh. **B**. Mosquitoes inside the container were allowed to feed on the solution for saliva collection via the artificial feeder. **C**. To measure saliva protein concentration, the protein concentrations of saliva that was collected from 1, 10, 20, 50, 100, and 150 female mosquitoes were examined using Bradford protein assays. Data were pooled from three independent experiments and displayed as mean ± SEM. nd, not detected; ***$P < 0.001$ (Unpaired $t$-test).

times every 2-h to obtain a larger volume of saliva-containing meal. The saliva-containing meal was sterilized via passage through an Ultrafree-GV centrifugal filter 0.22 μm before transfer into a new microtube. The sterilized saliva-containing meal was concentrated through a Millipore column with filter pore size 3 kDa nominal molecular weight limit (NMWL) and centrifuged at 12,000 $g$ for 1-h at 4˚C. The solution flowing through was discarded, and the remaining solution was collected and stored at -80˚C for further use.

### Plaque assay analysis

A pool of infected *Ae. aegypti* saliva was collected at 3, 5, 7, 10, and 14 days post infection via the artificial feeder. The saliva from individual day post infection was adjusted to an equivalent protein concentration before virus titer determination by plaque assay. C6/36 cells were used for plaque assays as described previously [13, 24]. Cell monolayers were rinsed with PBS, and 200 μl of infectious mosquito saliva was added to the cell monolayers for 2-h absorption at 28˚C. After absorption, 500 μl of 1% methylcellulose cell media was added, and the plates were maintained in the incubator at 28˚C. After 5 days, plates were fixed with 4% formaldehyde for 1-h at room temperature. Methylcellulose overlays were then removed, and plates were stained with 1% crystal violet for 1-h.

### Statistical analysis

The statistical analyses were performed using GraphPad Prism 5 software. Differences between groups were evaluated with unpaired *t*-tests. A *P*-value < 0.05 was considered to be statistically significant.

### Results and discussion

Feeding efficiency is a relevant parameter that needs to be considered in the development of artificial feeders [2, 11]. To evaluate if there is better feeding efficiency of the parafilm artificial blood feeder compared to live animals, *Ae. aegypti* and *Ae. albopictus* were allowed to feed on our artificial blood feeder and mice for the same period (Fig 1A and 1B). The total number of engorged females was counted to compare the efficiency between the two different blood sources (Fig 1C and 1D and S1 Table). In *Ae. aegypti*, an average of 84.67% females that were fed on artificial blood feeder exhibited engorgement compared to 60% of females fed on mice (S1 Table). Furthermore, there was a significant difference in the mean percentage of engorged females between these two groups (*P* < 0.05, Fig 1C). However, there was not a significant difference in the mean percentage of engorged *Ae. albopictus* (*P* > 0.05, Fig 1D). An average of 87.33% females fed on the artificial blood feeder using sheep erythrocytes exhibited engorgement compared to 85.33% of females fed on mice (S1 Table). The lack of a significant difference in the mean percentage of engorged females between these two groups may be because *Ae. albopictus* feeds on a variety of mammals such as swine, sheep, dogs, cats, and mice [28]. In both *Ae. aegypti* and *Ae. albopictus* there was no significant difference in the number of eggs laid between the individual females fed on artificial blood feeder and mice (*P* > 0.05, Fig 1E and 1F and S2 Table). Therefore, these results demonstrate that our artificial blood feeder provides as efficient a blood source as mice for mosquitoes.

We suggest that the artificial feeder could be used to replace live animals for female mosquito blood feeding. The results of a simple artificial membrane-feeding method using a standard conical tube and Parafilm-M membrane reported previously did not differ significantly in the numbers of mosquitoes compared with those using the mice-feeding method [11]. Our results showed a significantly higher percentage of mosquitoes (*Ae. aegypti*) fed on our device than fed on live mice, demonstrating that the development of a simple artificial membrane-

feeding method using Parafilm M membrane was successful. However, it is difficult to compare the feeding efficiency among different artificial feeders because of great variations in experimental conditions, such as different blood sources for the control and experimental groups. The preparation of blood sources before loading to Parafilm was also reported to be critical [11]. Moreover, the available surface of the mouse for biting by mosquito females in the control group was limited to 2 cm², the same area as the artificial blood feeder. A higher level of feeding efficiency was observed when using the whole mouse body rather than a 2 cm² area.

This artificial feeder can also serve as a useful apparatus for mosquito saliva collection. To collect saliva, *Ae. aegypti* and *Ae. albopictus* females were allowed to feed on the meal through the artificial feeder (Fig 2A and 2B). Our results revealed that the amount of protein detected in the artificial feeding solution increased according to the number of *Ae. aegypti* and *Ae. albopictus* that were engorged after exposure to the device ($P < 0.001$, Fig 2C and 2D and S3 Table). However, using the artificial feeder to collect sufficient saliva for evaluating protein concentration required at least 20 females fed on 100 μl of the feeding solution in this device (Fig 2C and 2D and S3 Table). This result suggests that our method is not suitable for collecting saliva from individual mosquitoes, but it enables the collection of saliva from multiple mosquitoes at once.

For infection with virus, *Ae. aegypti* consumed infectious blood using our artificial feeder loaded with washed sheep erythrocytes mixed with C6/36 cultured DENV2. We collected and evaluated DENV infection in mosquitoes' saliva on days 3, 5, 7, 10 and 14 post infectious blood meal. From 81% to 84% of mosquitoes that fed on the saliva collection device reached full engorgement (S4 Table). Our results showed that the plaque-forming units of infected *Ae. aegypti* saliva could be detected from day 7 after infection (Fig 3A and 3B). Although the time post infectious blood meal to collect saliva by artificial blood feeder could increase infected saliva titers, no significant differences were observed between day 7 and 10 post infectious blood meal ($P > 0.05$, Fig 3B). However, significant differences were observed between day 7 and 14 post infectious blood meal, and the highest titer was found on day 14 after infectious blood meal ($P < 0.05$, Fig 3B). The results of DENV transmission, using our method for infected saliva collection, show that virus can be detected as early as 7 days after oral infection. During days 7 to 14, there is an increase in virus titers, but the mosquito preference to feed on our device is unchanged (S4 Table).

Using our method, the collected saliva had sufficient virus titer for studying DENV transmission by plaque assay, or by using a mouse model to determine hemorrhage development [24]. These indicate that the artificial blood feeder is an efficient method to collect DENV2-infected mosquito saliva for virus transmission study. Virus transmission using region-delimited mosquito biting in mouse ears has been studied [29–30]. However, these virus transmission studies could not control the amount of saliva that the mosquito injected into the mouse, and required a high titer of infected saliva to investigate mice hemorrhage development. It is clear that methods to collect infected mosquito saliva should be developed. There is a previous study used this parafilm artificial feeder to collect DENV2-infected mosquito saliva for tests on mice, but the process and efficiency of this method were not described [24].

A previous study used a suspension feeding method to collect saliva by allowing the mosquitos to feed from a hanging blood droplet that was placed on the top of the mosquito cage, and then the droplet was tested for the presence of the virus [16]. There is a mosquito salivation device that was constructed with plexiglass as the surface on which mosquitoes were placed, and immersion oil was added to capillary tubes with a sterile pipette tip inserted into the opposite end [21]. The droplet feeding method demonstrated that virus transmission rates were consistent with the capillary tube method [31], but these saliva collection systems were

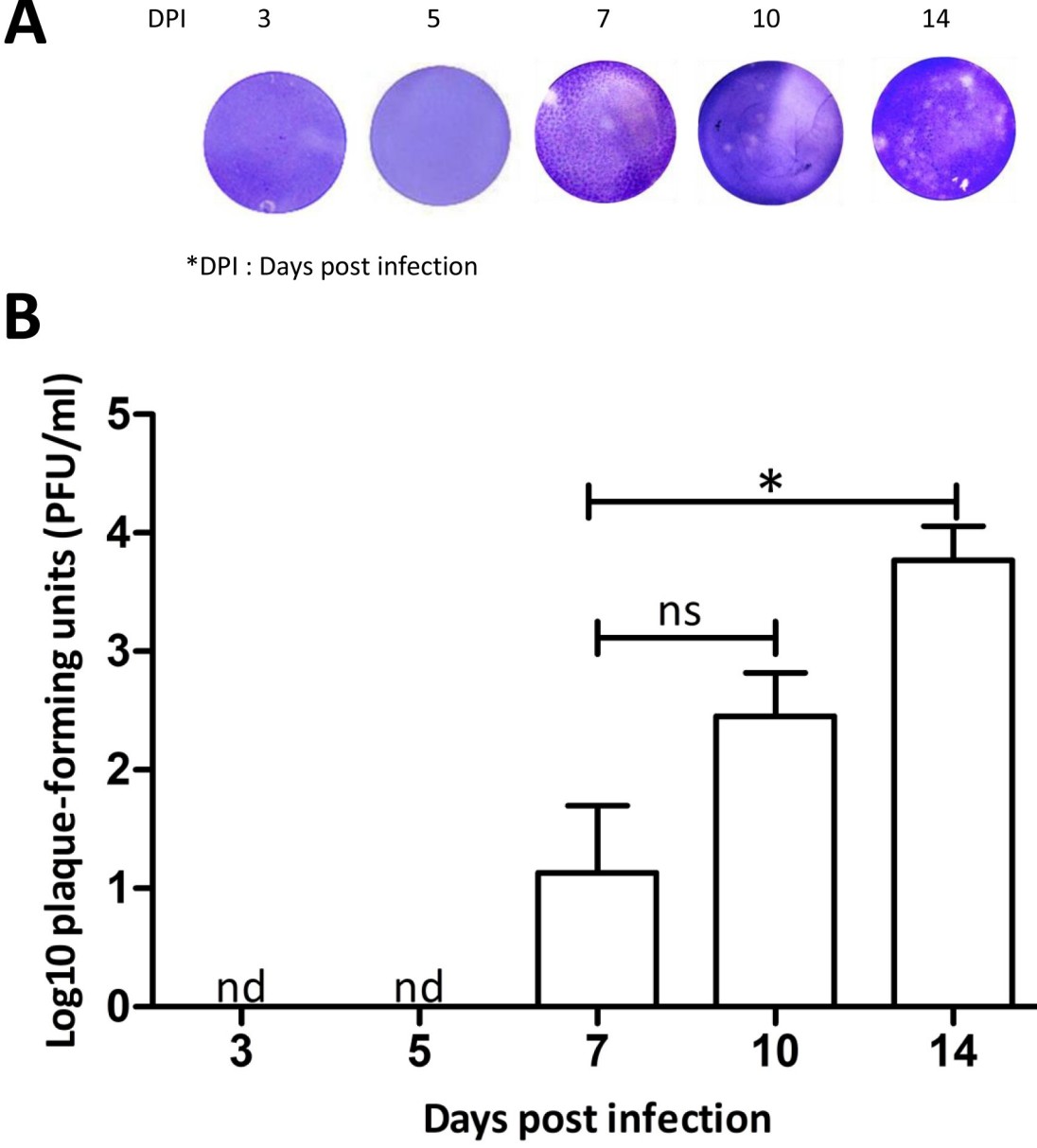

**Fig 3. Evaluation of saliva titer collected by artificial blood feeder.** DENV2-infected mosquito saliva was collected at 3, 5, 7, 10, and 14 days post infection via the artificial feeder. The saliva from individual day post infection was adjusted to an equivalent protein concentration. **A**. Comparison of plaques produced by DENV2-infected mosquito saliva at different days post infection. The infected saliva was diluted 10 fold and assayed in C6/36 monolayer cells. **B**. DENV2-infected mosquito saliva titer at different days post infection was pooled from three independent experiments and presented as mean ± SEM. Total 200 mosquitoes were used for each replicate. nd, not detected; ns, not significant; $^*P < 0.05$ (Unpaired $t$-test).

based on experiments involving small sample sizes ($N < 150$) [16, 21, 31]. In another experiment using the midge, *Culicoides nubeculosus*, saliva was collected from more than 5,000 *C. nubeculosus* in 20 pill boxes using a modified artificial membrane feeding system in which midges, attracted by 37°C horse blood, deposited saliva proteins in the Durapore filters [32]. However, this method used inner and outer glass chambers made of costly materials and complicated to assemble. Our device provides a new simple method of mosquito saliva collection

that is cost effective and useful for large sample sizes (150–200 mosquitoes in a single container). Moreover, we can increase the sample sizes by collecting mosquito saliva from many containers depending on requirements. Our experiments indicated that mosquito saliva titers and salivary protein concentrations that we collected are higher than those measured in previous studies [17–21].

This artificial feeder does not require special materials or electric heaters. The defined prewarmed temperature for blood mimics vertebrate blood temperature [11], and the meal contained FBS and ATP as a phagostimulant [33]. The artificial blood feeder specified here is inexpensive and easy to assemble with easily available common materials. Therefore, it can be considered as an alternative for blood sources from live animals to maintain mosquito populations in the laboratory. Furthermore, it is useful for oral infection, saliva collection from multiple mosquitoes at once, and studying dengue virus transmission in vector mosquitoes.

## Supporting information

**S1 Table. Blood-feeding efficiency of artificial blood feeder compared with mice blood-fed female mosquitoes.**
(DOCX)

**S2 Table. Eggs laid efficiency of individual female mosquito that fed on artificial blood feeder or mice blood-fed.**
(DOCX)

**S3 Table. Comparison of saliva protein concentration that collected from different number of mosquitoes using artificial feeder.**
(DOCX)

**S4 Table. Comparison of saliva titers collected from DENV2-infected *Ae. aegypti* via artificial feeder at different days post infection.**
(DOCX)

## Acknowledgments

The authors thank Prof. Roger F. Hou, National Chung Hsing University, for critical reading of the manuscript.

## Author Contributions

**Conceptualization:** Chalida Sri-in, Shih-Che Weng, Wu-Chun Tu.

**Data curation:** Chalida Sri-in.

**Funding acquisition:** Wu-Chun Tu.

**Investigation:** Chalida Sri-in.

**Methodology:** Shih-Che Weng, Wu-Chun Tu.

**Project administration:** Wu-Chun Tu.

**Resources:** Wu-Chun Tu.

**Supervision:** Shin-Hong Shiao, Wu-Chun Tu.

**Visualization:** Chalida Sri-in.

**Writing – original draft:** Chalida Sri-in.

**Writing – review & editing:** Wu-Chun Tu.

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
