## [Decision Letter · Decision Letter 0]

26 Feb 2020

PONE-D-19-36024

A simplified method for blood feeding, oral infection, and saliva collection of the dengue vector mosquitoes

PLOS ONE

Dear Dr Tu,

Thank you for submitting your manuscript to PLOS ONE. After careful consideration, we feel that it has merit but does not fully meet PLOS ONE’s publication criteria as it currently stands. Therefore, we invite you to submit a revised version of the manuscript that addresses the points raised during the review process.

Please carefully check all the aspects both reviewers have raised around your manuscript, specially related to the writing. A thorough grammar review should be performed in the end after all other points have been addressed.

We would appreciate receiving your revised manuscript by APRIL 10th. To enhance the reproducibility of your results, we recommend that if applicable you deposit your laboratory protocols in protocols.io, where a protocol can be assigned its own identifier (DOI) such that it can be cited independently in the future. For instructions see: http://journals.plos.org/plosone/s/submission-guidelines#loc-laboratory-protocols

We look forward to receiving your revised manuscript.

Kind regards,

Luciano Andrade Moreira, PhD

Academic Editor

PLOS ONE

Journal Requirements:

"This study was supported by research grant (MOST 107-2313-B-005-027) from Ministry of Science and Technology, Executive Yuan, ROC. The authors thank Prof. Roger F. Hou, National Chung Hsing University, for critical reading of the manuscript."

"No, the funders had no role in study design, data collection and analysis, decision to publish, or preparation of the manuscript. "

3. Thank you for stating the following in the Methods Section of your manuscript:

"The mouse strain BALB/c was a generous from Lexco Biotechnology Co., Ltd"."

"No, the funders had no role in study design, data collection and analysis, decision to publish, or preparation of the manuscript. "

Additionally, because some of your funding information pertains to commercial funding, we ask you to provide an updated Competing Interests statement, declaring all sources of commercial funding.

In your Competing Interests statement, please confirm that your commercial funding does not alter your adherence to PLOS ONE Editorial policies and criteria by including the following statement: "This does not alter our adherence to PLOS ONE policies on sharing data and materials.” as detailed online in our guide for authors  http://journals.plos.org/plosone/s/competing-interests.  If this statement is not true and your adherence to PLOS policies on sharing data and materials is altered, please explain how.

Please include the updated Competing Interests Statement and Funding Statement in your cover letter. We will change the online submission form on your behalf.

Reviewers' comments:

Reviewer's Responses to Questions

**Comments to the Author**

1. Is the manuscript technically sound, and do the data support the conclusions?

Reviewer #1: Yes

Reviewer #2: Yes

2. Has the statistical analysis been performed appropriately and rigorously? 

Reviewer #1: Yes

Reviewer #2: Yes

3. Have the authors made all data underlying the findings in their manuscript fully available?

Reviewer #1: Yes

Reviewer #2: Yes

4. Is the manuscript presented in an intelligible fashion and written in standard English?

Reviewer #1: No

Reviewer #2: No

5. Review Comments to the Author

Reviewer #1: In this manuscript, Sri-in et al propose a simple and elegant method for mosquito feeding under laboratory conditions. Different strategies, methods, and apparatus for mosquito feeding have been proposed along the past years, but usually they require a heat source or other labware. Here, the authors propose a simplified system using stretched parafilm to offer a blood or protein meal to mosquitoes. Their findings suggest that using wrapped blood inside parafilm is comparable to using mice as blood source for mosquito feeding regarding the number of engorged females and number of eggs laid per female. Also, it seems possible to recover saliva proteins and virus expelled in the saliva by infected mosquitoes using their proposed feeding system, enabling transmission experiments to be done in a simplified manner. Thus, this reviewer recommends the paper for publication with minor revision. Listed below are misunderstandings that must be fixed before publication, together with necessary clarification in methods and statistics. This reviewer will feel pleased to review the rebuttal of this manuscript upon resubmission.

Major comments:

Line 69-71 – […] “saliva collected does not contain sufficient virus titers for studying DENV transmission by means of plaque assay or mouse model.” […] – This affirmation is not entirely true. It is difficult to detect infected saliva from infected mosquitoes but is not impossible. Different studies have studied virus transmission using region-delimited mosquito biting in mouse ears (e.g. Secundino NFC et al, 2017 – doi 10.1186/s13071-017-2286-2 or Cox J et al, 2012 – doi 10.1128/JVI.00534-12). In fact, it is discussed later on in the text that their method is only suitable for collection of saliva from pools of at least 20 mosquitoes (Fig 2 C/D – Lines 210-214) thus a comparison with other methods should be discussed in the text.

Line 219-232 – Would have the authors verified the prevalence of DENV infection in individual mosquitoes upon feeding on their system comparing to a water-jacked or other previously characterized system? Their assessment of saliva detection of DENV was done in large pools of mosquitoes but could only represent a very low percentage of mosquitoes that were infected, thus representing that their system may not be efficient to infect mosquitoes. The authors even emphasize that their results “indicated that the artificial blood feeder is an efficient artificial feeder” (lines 231-232) and their results do not hold sufficient information for this affirmation. Also, it is interesting to include more details (e.g. size of mosquito pool) of the saliva DENV-detection experiment in the figure legend.

Minor comments:

For a better organization of the paper, I suggest moving Tables to supplementary information.

Line 66 – Break into a new paragraph. Ideas are not connected with previous lines.

Line 98 – Please add sentence explaining that plaque assay method will be further explained in the text.

Line 100 – Invert “laying eggs” by “egg laying”

Line 148 – “The infected mosquitoes were allowed to salivate for 30 min.” – Mosquitoes did not salivate; they were allowed to bite the candy wrap. Please rephrase.

Line 150 – “5 times” not “5 time”

Line 154 – Describe better “Millipore column measuring 3 kDa”

Line 159-160 – Is it correct that plaque assay was done using C6/36 cells? Usually these cells do not form plaques upon DENV infection, thus should not be possible to visualize using crystal violet. Please clarify.

Line 173 – change “elucidate” by “evaluate if there is”

Line 175 – “…albopictus were fed on” by “…albopictus were allowed to feed on”

Line 178 – Was it considered fully engorged females or mosquitoes that ingested any amount of blood? Please clarify.

Line 183-184 – “This result may …” This sentence brings no relevant information. Please rephrase.

Line 193 and 198 – Is reference 11 correct here?

Line 193-196 – Please clarify that only for Ae. aegypti you found significantly higher percentage of mosquitoes fed on parafilm candy.

Line 199 – The discussion starting by “In addition” is important but is lacking context on the way it’s written. Please explain or rephrase.

Line 208 – “engorged” not “engorge”

Line 215-216 – Please rephrase

Line 230 – What is high virus concentration?

Line 230-231 – Explain the sentence “Interestingly, the infected saliva collection…”

Line 233 – Lost discussion in the text. Please include context and rephrase.

Table 4 – What means “a” along the statistics represented in this table?

Reviewer #2: I feel that this is good paper describing a low cost alternative to standard artificial blood-feeder devices that are commercially available. The finding presented here validate this new method as an acceptable alternative, I believe these findings will have a great impact on the ability for labs in lower income countries to be able to perform more complex vector competence and transmission studies.

The only issue that I have found with this article is language based, there are several instances throughout the paper in which the phrasing used is confusing and hard to understand the point being made. I have highlighted these sections in my review of the article and believe that the correction of these minor issues will make for a good article.

6. PLOS authors have the option to publish the peer review history of their article (what does this mean?). If published, this will include your full peer review and any attached files.

Reviewer #1: No

Reviewer #2: No

---

## [Author Response · Author response to Decision Letter 0]

13 Apr 2020

Respond to Reviewers’ Comments to the Authors

Reviewer #1: 

In this manuscript, Sri-in et al propose a simple and elegant method for mosquito feeding under laboratory conditions. Different strategies, methods, and apparatus for mosquito feeding have been proposed along the past years, but usually they require a heat source or other labware. Here, the authors propose a simplified system using stretched parafilm to offer a blood or protein meal to mosquitoes. Their findings suggest that using wrapped blood inside parafilm is comparable to using mice as blood source for mosquito feeding regarding the number of engorged females and number of eggs laid per female. Also, it seems possible to recover saliva proteins and virus expelled in the saliva by infected mosquitoes using their proposed feeding system, enabling transmission experiments to be done in a simplified manner. Thus, this reviewer recommends the paper for publication with minor revision. Listed below are misunderstandings that must be fixed before publication, together with necessary clarification in methods and statistics. This reviewer will feel pleased to review the rebuttal of this manuscript upon resubmission.

Major comments:

Line 69-71 – […] “saliva collected does not contain sufficient virus titers for studying DENV transmission by means of plaque assay or mouse model.” […] – This affirmation is not entirely true. It is difficult to detect infected saliva from infected mosquitoes but is not impossible. Different studies have studied virus transmission using region-delimited mosquito biting in mouse ears (Cox J et al, 2012, NFC et al, 2017). In fact, it is discussed later on in the text that their method is only suitable for collection of saliva from pools of at least 20 mosquitoes (Fig 2 C/D – Lines 210-214) thus a comparison with other methods should be discussed in the text.

Ans: We took the reviewer’s suggestion to revise our manuscript. However, our purpose is to collect sufficient infected saliva to study virus transmission such as plaque assay and mice model (Sri-in et al., 2019). Using the parafilm candy method, we could collect sufficient infected saliva for any mosquito-borne virus transmission studies and could adjust an equivalent saliva protein concentration of every experimental sample to avoid biased results. As reviewer’s suggestion, a comparison with other methods was discussed in the text lines 224-231.”

Therefore, this paper was provided to describe a new method as an acceptable alternative to standard artificial blood-feeder and saliva collection devices that are commercially available. We believe these findings will have a helpful impact on the ability for labs to be able to perform more complex vector competence and transmission studies. 

Sri-in C, Weng SC, Chen WY, Wu-hsieh BA, Tu WC, Shiao SH. A salivary protein of Aedes aegypti promotes dengue-2 virus replication and transmission. Insect Biochem Mol Biol 2019;111(May): 103181. https://doi.org/10.1016/j.ibmb.2019.103181

Line 219-232 – Would have the authors verified the prevalence of DENV infection in individual mosquitoes upon feeding on their system comparing to a water-jacked or other previously characterized system? Their assessment of saliva detection of DENV was done in large pools of mosquitoes but could only represent a very low percentage of mosquitoes that were infected, thus representing that their system may not be efficient to infect mosquitoes. The authors even emphasize that their results “indicated that the artificial blood feeder is an efficient artificial feeder” (lines 231-232) and their results do not hold sufficient information for this affirmation. Also, it is interesting to include more details (e.g. size of mosquito pool) of the saliva DENV-detection experiment in the figure legend.

Ans: We added size of mosquito pool that used in this experiment in the figure legend (line 384) and table S4. Thank you the reviewers for letting us rethink this issue, especially regarding the inability to know the individual virus content in a pool of mosquitoes, or to accurately calculate the virus power of a single mosquito individual. Our purpose is to collect the saliva from vector mosquitoes that were infected by oral infection. We had verified the saliva from individual mosquito that infected by oral infection but the saliva titer is not high enough to estimate by plaque assay. Therefore, the assessment of saliva detection of DENV was done in large pools of mosquitoes. Moreover, we had already mentioned that our method is not suitable for collecting saliva from individual mosquitoes, but it enables the collection of saliva from multiple mosquitoes at once (line 205-206). 

Collecting saliva from vector mosquitoes is not an easy task. It is even more difficult to collect saliva from vector mosquitoes containing dengue virus that were infected by oral infection. We succeed to collect saliva containing virus that show in figure 3 and table S4; the saliva was diluted 10 fold and estimate by plaque assay. This indicated that the artificial blood feeder is an efficient method for collected saliva containing virus (the sentence was corrected, see line 223-224). We understand as same as reviewer that this technique cannot prove the above things. The reason why this report conducts a plaque assay is just to show this method can indeed simulate the situation of real mosquito bites. The saliva containing virus collected by this method is not contaminated by salivary gland tissue or other mosquito tissues, and these dengue viruses still have infection activity. 

Secondly, our method does not require additional heat sources. In addition to the simplicity of the method, it may be helpful to prolong the time of virus infection activity. This is the advantage of using artificial methods to infect vector mosquitoes with mosquito-borne viruses.

Minor comments:

For a better organization of the paper, I suggest moving Tables to supplementary 

information.

Ans: Thank you for suggestion, this has been done.

Line 66 – Break into a new paragraph. Ideas are not connected with previous lines.

Ans: The text has been revised in accordance with the opinion of the reviewer. (Line 55)

Line 98 – Please add sentence explaining that plaque assay method will be further explained in the text.

Ans: Sentence added. (Line 89)

Line 100 – Invert “laying eggs” by “egg laying”

Ans: Corrected. (Line 91)

Line 148 – “The infected mosquitoes were allowed to salivate for 30 min.” – Mosquitoes did not salivate; they were allowed to bite the candy wrap. Please rephrase.

Ans: The sentence has been reworded. (Line 139)

Line 150 – “5 times” not “5 time”

Ans: Corrected. (Line 142)

Line 154 – Describe better “Millipore column measuring 3 kDa”

Ans: Description changed from “Millipore column measuring 3 kDa” to “Millipore column with filter pore size 3 kDa nominal molecular weight limit (NMWL)”. (Line 144-146)

Line 159-160 – Is it correct that plaque assay was done using C6/36 cells? Usually these cells do not form plaques upon DENV infection, thus should not be possible to visualize using crystal violet. Please clarify.

Ans: There are many laboratories that use BHK cells for viral titration, but C6/36 cells also can be used for plaque assay upon DENV infection using the protocol from these previous studies (Line 153): 

(1) Das S, Garver L, Ramirez JR, Xi Z, Dimopoulos G., 2008. Protocol for dengue infections in mosquitoes (A. aegypti) and infection phenotype determination. Journal of Visualized Experiments. (5): 4–5. 

(2) Sri-in C, Weng SC, Chen WY, Wu-hsieh BA, Tu WC, Shiao SH. A salivary protein of Aedes aegypti promotes dengue-2 virus replication and transmission. Insect Biochem Mol Biol 2019;111(May): 103181. https://doi.org/10.1016/j.ibmb.2019.103181

Line 173 – change “elucidate” by “evaluate if there is”

Ans: Changed. (Line 166)

Line 175 – “…albopictus were fed on” by “…albopictus were allowed to feed on”

Ans: Corrected. (Line 168)

Line 178 – Was it considered fully engorged females or mosquitoes that ingested any amount of blood? Please clarify.

Ans: Yes, in order to ensure that the female mosquitoes will take blood meal well, the female mosquitoes were starvation treatment before the experiment. When the experiment was conducted, we observed that female mosquitoes were full of blood after one minute of feeding. The explanation was added in line 104-105.

Line 183-184 – “This result may …” This sentence brings no relevant information. Please rephrase.

Ans: Thank you for reminder. The sentences have been rephrased, refer to the relevant lines of the text. (Line 177-179)

Line 193 and 198 – Is reference 11 correct here?

Ans: We are sorry for the mistake. The references were rearranged. (See References list and line 188-194). The correct reference as follow;

Costa-da-silva L, Navarrete R, Salvador FS, Karina-costa M, Rocha R, Capurro ML. Glytube : a conical tube and parafilm M-based method as a simplified device to artificially blood-feed the dengue vector mosquito , Aedes aegypti. PLoS One. 2013;8(1):1–5. 

Line 193-196 – Please clarify that only for Ae. aegypti you found significantly higher percentage of mosquitoes fed on parafilm candy.

Ans: This section has been clarified in text line 188-197.

A higher level of feeding efficiency with mice was observed when the whole body of mice rather than a 2 cm2 hole in the mouse body was used that is a mouse available surface for biting by mosquito females in the control group was limited to the same square centimeter of the artificial blood feeder. Moreover, the result that found significantly higher percentage of mosquitoes fed on parafilm candy is only for Ae. aegypti but not Ae. albopictus, may be because Ae. albopictus fed on a variety of mammals such as swine, sheep, dog, cat, and mice (line 178-179) that is different from Ae. aegypti. 

Line 199 – The discussion starting by “In addition” is important but is lacking context on the way it’s written. Please explain or rephrase.

Ans: The sentence has been rephrased. See line 191-193.

Line 208 – “engorged” not “engorge”

Ans: Corrected. (Line 201)

Line 215-216 – Please rephrase

Ans: The sentence has been rephrased, refer to the relevant lines of the text. (Line 207-208).

Line 230 – What is high virus concentration?

Ans: We changed the word “virus concentration” to “virus titer” and the sentence was rephrased. (Line 221).

Line 230-231 – Explain the sentence “Interestingly, the infected saliva collection…”

Ans: Sri-in et al. (2019) had ever used parafilm candy method to collect sufficient DENV2-infected mosquito saliva for the mice test to examine hemorrhage development, but the process and efficiency of this method were not described. The sentence has been rewritten. (Line 221-223 and 229-231).

 Sri-in C, Weng SC, Chen WY, Wu-hsieh BA, Tu WC, Shiao SH. A salivary protein of Aedes aegypti promotes dengue-2 virus replication and transmission. Insect Biochem Mol Biol 2019;111(May): 103181. https://doi.org/10.1016/j.ibmb.2019.103181

Line 233 – Lost discussion in the text. Please include context and rephrase.

Ans: This section has been rewritten. (Line 232)

Table 4 – What means “a” along the statistics represented in this table?

Ans: a, base values for comparison with other values; ab, not significant; b, P < 0.05. Explanation was added to Table S3 and S4.

Reviewer #2: 

I feel that this is good paper describing a low cost alternative to standard artificial blood-feeder devices that are commercially available. The finding presented here validate this new method as an acceptable alternative, I believe these findings will have a great impact on the ability for labs in lower income countries to be able to perform more complex vector competence and transmission studies.

The only issue that I have found with this article is language based, there are several instances throughout the paper in which the phrasing used is confusing and hard to understand the point being made. I have highlighted these sections in my review of the article and believe that the correction of these minor issues will make for a good article.

Lines 35-37 need some revising; the current sentence structure is complicated and confusing.

Ans: We have revised these lines. (Line 26-28)

Lines 40-43 needs to be revised; again, the current sentence structure is confusing and hard to follow.

Ans: We have revised these lines. (Lines 30-33)

Line 55 consider changing to read as; …feeding these vectors due to ethical approval, …

Ans: Corrected. (Line 44)

Line 57 might read better as; …using live animals is costly and maintaining them is laborious.

Ans: Corrected. (Line 46)

Line 58 should read; Artificial blood-feeding systems are important…

Ans: Corrected. (Line 47)

Line 60 consider revising as; …feed blood-sucking insects [6-7], most of them …

Ans: Corrected. (Line 49)

Lines 61-62 consider revising as; …thin membranes filled with warmed animal blood, that allow female mosquitoes to insert their proboscis to acquire a blood meal.

Ans: Corrected. (Line 50-51)

Lines 67-74 these sentences need revising

Ans: We have revised these lines 56-64.

Line 83 consider revising as; Mouse-use for blood …

Ans: Corrected. (Line 73)

Line 85 consider revising as; BALB/c mice were obtained …

Ans: Corrected. (Line 75)

Line 90 should read; …dose of 75mg/kg i.p. during the blood feeding experiment …

Ans: Corrected. (Line 80)

Line 100 might read better as; Mosquito blood-feeding and egg laying experiments

Ans: Corrected. (Line 91)

Line 111 the authors state that they determined if mosquitoes had taken a full meal and then calculated the percentage of engorged females. Could the authors further elaborate on what is meant by a full meal and the criteria used to determine this?

Ans: In order to ensure that the female mosquitoes will take blood meal well, the female mosquitoes were starvation treatment before the experiment. When the experiment was conducted, we observed that female mosquitoes were full of blood after one minute of feeding. The explanation was added in line 104-105.

Line 124 should read as; …washed sheep erythrocytes,…

Ans: Corrected. (Line 117)

Lines 126-127 this sentence is hard to follow, please consider rephrasing

Ans: Corrected. (Lines 118-120)

Lines 150-153 are hard to follow, please revise this section

Ans: We have revised this section. (Lines 142-144)

Line 153 consider changing to; The sterilized saliva-containing …

Ans: Changed as suggested. (Lines 144-145)

Line 158 unsure what is meant by this sentence, could the authors either give a brief explanation of what is meant or revise this sentence?

Ans: The sentence has been revised. (Lines 150-152). 

Line 174 change to; … artificial blood feeder compared to live animals, …

Ans: Changed as suggested. (Line 167)

Line 176 change to; …compare the efficiency between the two different blood …

Ans: Changed as suggested. (Lines 169-170)

Line 178 change to; …compared to 60% of females fed on …

Ans: Changed as suggested. (Line 171-172)

Lines 183-184 Could the authors explain what they are trying to state with this sentence? I am unsure if they mean that albopictus have a preference for human blood or if their preference is for other mammals?

Ans: The sentences have been corrected. (Line 178-179)

Line 190 should read; …for female mosquito blood feeding.

Ans: Corrected. (Line 184-185)

Lines 203-204 This sentence needs rephrasing

Ans: The sentences have been rephrased, refer to the relevant lines of the text. (Lines 194-197)

Line 208 should read; …the number of engorged Ae. aegypti…

Ans: Corrected. (Line 200-201)

Line 215 should read as; To infect with virus, Ae. aegypti consumed infectious blood using …

Ans: Corrected. (Line 207)

Lines 216-218 these sentences are hard to follow, please consider rephrasing

Ans: Corrected. (Line 208-210)

Line 229 should read; … we could collect sufficiently infected saliva…

Ans: We want to explain that the sufficient saliva containing virus will be collected when use our method. The sentence was corrected (Line 221-222)

Lines 244-246 Could the authors expound what is meant here, it is unclear to me? Are they describing limitations of other experiments are based on the sample size of mosquitoes used? Or implying that number of mosquitoes in a single container is the limiting factor?

Ans: Our device provides a new simple method of mosquito saliva collection that is cost effective and useful for large sample sizes in once (150-200 mosquitoes in a single container). Moreover, we can increase the sample sizes by collecting mosquito saliva from many containers at the same time depending on requirements. (Line 244-247)

Lines 321-322 this is confusing, please revise this sentence

Ans: We have revised these lines. (Line 355-357)

Line 345 should read; …saliva was collected at 3, 5, 7, 10, and 14 days post infection via…

Ans: Corrected. (Line 379)

Lines 347-348 should read; …DENV-2 infected mosquito saliva at different days post infection.

Ans: Corrected. (Line 381-382)

Line 349 should read; …mosquito saliva titer at different days post infection…

Ans: Corrected. (Line 383)

---

## [Decision Letter · Decision Letter 1]

11 May 2020

A simplified method for blood feeding, oral infection, and saliva collection of the dengue vector mosquitoes

PONE-D-19-36024R1

Dear Dr. Tu,

We are pleased to inform you that your manuscript has been judged scientifically suitable for publication and will be formally accepted for publication once it complies with all outstanding technical requirements.

Please check that are still some comments from both reviewers which might improve your final manuscript.

With kind regards,

Luciano Andrade Moreira, PhD

Academic Editor

PLOS ONE

Additional Editor Comments (optional):

Reviewers' comments:

Reviewer's Responses to Questions

**Comments to the Author**

1. If the authors have adequately addressed your comments raised in a previous round of review and you feel that this manuscript is now acceptable for publication, you may indicate that here to bypass the “Comments to the Author” section, enter your conflict of interest statement in the “Confidential to Editor” section, and submit your "Accept" recommendation.

Reviewer #1: All comments have been addressed

Reviewer #2: (No Response)

2. Is the manuscript technically sound, and do the data support the conclusions?

Reviewer #1: Yes

Reviewer #2: Yes

3. Has the statistical analysis been performed appropriately and rigorously? 

Reviewer #1: Yes

Reviewer #2: Yes

4. Have the authors made all data underlying the findings in their manuscript fully available?

Reviewer #1: Yes

Reviewer #2: Yes

5. Is the manuscript presented in an intelligible fashion and written in standard English?

Reviewer #1: Yes

Reviewer #2: Yes

6. Review Comments to the Author

Reviewer #1: The authors addressed reviewers’ comments, and the paper is sound for publication according to this reviewer. Of note, the revised manuscript with Track Changes did not match the final PDF version, which made reviewing a complicated process.

Some minor suggestions to improve clarity in the final text:

Line 200-202 – “Our results revealed that the number of engorged Ae. aegypti and Ae. albopictus fed using the artificial feeder showed significantly increased protein concentration (…)”. Suggestion: “Our results revealed that the amount of protein detected in the artificial feeding solution increased according to the number of Ae. aegypti and Ae. albopictus that were engorged after exposure to the device (…)”.

Line 208-209 – “The most appropriate time to collect infected mosquito saliva was examined on day 3, 5, 7, 10 and 14 post infectious blood meal.” – Suggestion: “We collected and evaluated DENV infection in mosquitoes’ saliva on days 3, 5, 7, 10 and 14 post infectious blood meal.”

Line 211 – “…device engorged” to “…device reached full engorgement”.

Line 221-222 – “Using our method, the saliva was collected sufficient virus titer for studying DENV transmission by plaque assay (…)”. Suggestion: “Using our method, the collected saliva had sufficient virus titer for studying DENV transmission by plaque assay (…)”.

Reviewer #2: I still feel that this paper describing a novel low cost artificial blood-feeder is good and merits publication. The authors present methods utilized to validate this alternative blood feeding system, which has the potential to greatly impact the ability for labs in lower income countries to be able to perform more complex vector competence and transmission studies.

I have found this version to be much improved compared to the original submission, with much clearer language which it made it easier to follow. I did have a few minor critiques for this version which are not critical, however there were a couple of spots that I felt needed a little more clarification to make this a more polished article. With that said I look forward to this article to be published and utilized by labs with limited financial support.

7. PLOS authors have the option to publish the peer review history of their article (what does this mean?). If published, this will include your full peer review and any attached files.

Reviewer #1: No

Reviewer #2: No

---

## [Editor Report · Acceptance letter]

14 May 2020

PONE-D-19-36024R1 

A simplified method for blood feeding, oral infection, and saliva collection of the dengue vector mosquitoes 

Dear Dr. Tu:

I am pleased to inform you that your manuscript has been deemed suitable for publication in PLOS ONE. Congratulations! Your manuscript is now with our production department. 

With kind regards,

on behalf of

Dr. Luciano Andrade Moreira 

Academic Editor

PLOS ONE